# OpenReview forum: "GameBench: Evaluating Strategic Reasoning Abilities of LLM Agents"
_NeurIPS.cc/2024/Datasets_and_Benchmarks_Track — Submitted to NeurIPS 2024 Track Datasets and Benchmarks_

### Official Review · Reviewer_o93U · 2024-07-04
**Review of Submission 2214**

**Rating:** 3
**Confidence:** 4
**Correctness:** Seems correct
**Clarity:** Bad

**Review:**

Although it would indeed be beneficial for language model researchers to understand how the models perform under out-of-distribution scenarios when used as agents, this article, or the proposed GameBench, does not present much new information or insights on this topic. The major reasons are as follows:

1. Some highly related works are missing. There are many logic puzzle-solving works demonstrated in a recent survey [1], and some of the works have motivations and approaches that are very similar to GameBench. For example, [2] also emphasizes the "OOD task" for LLM's reasoning ability evaluation, and reached a conclusion similar to this article. The authors should give a thorough study of such works and sufficiently discuss the similarities and differences between these works and GameBench.

2. The narration is somewhat hard to follow. This is a benchmark paper focusing on the games that evaluate LLMs, but the games themselves are not even introduced in the article body! I understand it is hard to give a full description of the games, but the authors should at least put some one-line summary of the game, and how the games fall into the "reasoning categories" in Table 1. Again, the "reasoning categories" are not well-introduced either. Due to the missing of such information, it is very hard to interpret or gain any insights from the results.

3. The analysis of the results is insufficient. For instance, it is unclear when LLMs underperform, what the potential reasons might be, and what solutions could be proposed. You may also refer to [2] for result analysis, although their conclusions also lack concrete evidence and rely on guesswork.

[1] Giadikiaroglou, Panagiotis, et al. "Puzzle Solving using Reasoning of Large Language Models: A Survey." arXiv preprint arXiv:2402.11291 (2024).
[2] Li, Yinghao, Haorui Wang, and Chao Zhang. "Assessing logical puzzle solving in large language models: Insights from a minesweeper case study." arXiv preprint arXiv:2311.07387 (2023).

**Strengths:**

See "Review"

**Additional Feedback:**

N/A

**Documentation:**

Discussion is missing on how data is collected.

**Ethics:**

No.

**Limitations:**

See "Review"

**Opportunities For Improvement:**

- A group of citations should be put into one \citep{} block, e.g, \citep{schick2023toolformer, agent_gpt, auto_gpt}.
- It is not suggested to refer to GPT-3.5-* as GPT-3.
- Quotation marks are incorrect. Check the LaTeX guide.

**Relation To Prior Work:**

Highly related works missing

**Summary And Contributions:**

This paper proposes GameBench, a collection of 9 out-of-distribution game benchmarks that evaluate language model agents' strategic reasoning abilities. Experiments show that LLM underperforms human beings even when augmented with CoT or RaP techniques.

---

> ### Author Rebuttal · Authors · 2024-08-18
>
> We would like to thank the reviewer for their feedback and for introducing us to related works that are highly relevant.
>
> We will address some of your comments specifically and then make a broader statement.
> “For instance, it is unclear when LLMs underperform” – The benchmark is not designed to make an absolute statement about any agent’s performance, but is to compare performance between generations of agents and hopefully humans when more human data is collected.
> “what solutions could be proposed” – Our project was not designed around the improvement of LLM agents, which we discuss in the hazards section on line 464.
>
> “Analysis of results in insufficient, although their conclusions also lack concrete evidence and rely on guesswork” – Here is our general statement. We would like to clarify that the main contribution of our submission is the benchmark. We believe that the conclusions we make are within the bounds of evidence. And yes, we agree the size of our data is very limited, and so our conclusions are not very confident. Our analysis is meant as a starting point of possible interesting and important hypotheses that our limited data could be beginning to suggest, but would require further data collection and scientific investigation to confirm. Again, our main contribution is the benchmark, and the data is meant to demonstrate the potential of the benchmark rather than us having to discuss hypothetical results, which wouldn’t make for an interesting benchmark that others would want to use. We would’ve loved to collect more data and make stronger conclusions, but data collection was very cost-prohibitive for us.
>
> R3 – Thank you for these related works. We have analyzed them and here we present our major differences and similarities.
> Giadikiaroglou et al. looks specifically at puzzle games, whereas we are trying to capture the abstract concept of strategic reasoning which extends beyond just puzzle games. It appears their taxonomy does extend to deterministic and stochastic games, but the games they look at are certainly in-distribution (chess, sudoku, minesweeper, and poker). They are also looking at different prompting techniques, but they are not looking at the state-of-the-art prompting techniques, rather extensions of chain-of-thought (tree-of-thoughts, tree-of-uncertain-thoughts, graph-of-thoughts, etc.). We selected RAP not only for being SOTA, but also for its being optimized for game-based reasoning in order to demonstrate an upper-bound on frontier LLM reasoning performance, whereas these other prompting techniques are more general and would fall below this upper-bound.
> Li et al. looks only at a single game. They make sure this game is OOD which is what we try to do, but they draw conclusions about LLM reasoning abilities from performance on just this game. We understand that reasoning is multifaceted, which is why we implemented a diversity of games (with more to be added in the future). Additionally, our diversity of games ensures that performance is not due to inductive biases on any particular game being more easy or difficult to work with (e.g. maybe the format of a game just happens to be more difficult to understand). We are interested in the way they presented Minesweeper, a game that certainly has plenty of strategy guides online, in such a way that made it nonetheless out-of-distribution to an LLM.
>
> R3 – “games themselves are not even introduced in the article body” This was a tough decision for us to make as we didn’t want to break the flow of the paper by describing each individual game, which is why we relegated their descriptions to line 570. We imagined a reader would be satisfied with the general descriptions of the games’ themes in Table 1 to understand that their breadth, knowing that more specifics could be found in the appendix.
>
> R3 – “Again, the "reasoning categories" are not well-introduced either”: Our main goal was to emphasize that our selection has breadth. We feel this is an unfortunate consequence of strategic reasoning being a hard-to-dichotomize concept. This is why we chose categories that were as tangible as possible: cooperation – players working together; social deduction – players modeling other players’ decisions; language communication – players using natural language to interact; hidden information – players not knowing other players’ states; non-deterministic – some results left to chance; abstract strategy – requiring tree-based calculation.
>
> R3 – Latex comments – Also, thank you for these. We have incorporated them into our working version.

---

> > ### Comment · Reviewer_o93U · 2024-08-20
> > **Thanks for your detailed response**
> >
> > First I want to clarify that "conclusions lacking concrete evidence" is a comment to [2] rather than this work.
> >
> > I understand and agree that introducing the games is the first priority of a benchmark paper rather than discussing the performance of models. But unfortunately, I find that your article does not address either case very well. If the authors want to focus on the games, then a more detailed introduction to the games, followed by a discussion on the similarities and differences of the proposed games should be included in the article body. In addition, a more comprehensive comparison to previous works should also be included. If the authors believe an evaluation with the SOTA reasoning models and techniques is necessary, it is suggested to make the results more informative.
> >
> > As the authors proposed almost no improvement to the manuscript, I'll keep my current score.

---

> > > ### Author Rebuttal · Authors · 2024-08-21
> > >
> > > R3 – We wish to thank the reviewer for their prompt response to our rebuttal.
> > >
> > > R3 – Thank you as well for the clarifying comment.
> > >
> > > R3 – “...does not address either case very well.” To address one point at a time, we are unclear if you are suggesting to make changes to the paper relating to the performance of models. This comment suggests analysis may be lacking, although outside of the clarifying comment the response does not focus on this aspect.
> > >
> > > R3 – “If the authors want to focus on the games, then a more detailed introduction to the games...should be included in the article body.”  Thank you for mentioning the importance of including the game information in the main article. As we mentioned previously, we were unsure where to include this info and also agree with how a lack of details could create a lack of understanding of the benchmark.
> > >
> > > To remedy this, we would update Section 3.2 with the following:
> > >
> > > - Definitions of each reasoning category (3.2.1)
> > > - Descriptions of the games (1-2 lines) (3.2.2, 1st paragraph)
> > > - Descriptions of how each game fits into respective reasoning categories, in addition to Table 1 for an overview of all games (3.2.2, 2nd paragraph)
> > >
> > >
> > > A partial, rough draft suggested improvement is as follows:
> > >
> > > 3.2.1 Reasoning Category Definitions
> > > We use the following categories to define all the games within our benchmark. We wanted to cover a large spread of reasoning types to test different areas that would quantify strategic reasoning. This can be a difficult task to undergo and therefore this selection is not meant to be exhaustive.
> > >
> > > - Abstract Strategy: A model must make decisions likely involving the use of logical deduction from a game's rule sets.
> > > - Hidden information: A model doesn’t know everything about the game-state.
> > > - Language communication: A model must negotiate or otherwise communicate with the other model or player.
> > > - Cooperation: Each model can cooperate with a copy of itself. The copy has separate state and can only communicate with the model through in-game channels.
> > > - Non-Deterministic: A model will interact with events from the game that cannot be planned for in advance.
> > > - Social Deduction: A model will need to determine the secret status of another, actively disguise their own status, or both.
> > >
> > > 3.2.2 Category and Game Matching
> > >
> > > Air, Land, and Sea is a war strategy game where players are Supreme Commanders fighting to
> > > control two of three areas (air, land, sea) by deploying limited Battle card forces each round. The first
> > > commander to accumulate 12 points across multiple battles wins the war. [1] Full rule sets are included in Appendix G.
> > >
> > > [1] https://boardgamegeek.com/boardgame/247367/air-land-and-sea
> > >
> > > Air, Land, and Sea falls under the category of Abstratic Strategy as the model has three different avenues to decide to play in, as well as different card orderings that can be performed. From the rule book, "The order in which you play your cards is critical...", and thus gives the model a degree of freedom for creating abstract strategy. It also falls under Non-Determinism as the players are dealt cards randomly and have to base their strategy from these. (Then repeat for each game)
> > >
> > > [[end example]]
> > >
> > >
> > > A full length description of all games would be included in final manuscript. The above example is meant to quickly get feedback on whether this would help the reader through the paper in this format. Should this be the case, we would send a full section revision for updating our manuscript.
> > >
> > >
> > >
> > > R3 – “In addition, a more comprehensive comparison to previous works should also be included.”
> > >
> > > Our interpretation of this comment isn't certain, and if our response doesn't fully answer the statement please let us know where we missed and we will make sure to respond to that.
> > >
> > > What we did interpret was that this was in response to our comment "We selected RAP not only for being SOTA, but also for its being optimized for game-based reasoning..." and you would prefer us specify further this point in section 3.1 "Agent and scaffolding selection" as well as the related works section under "LLM agents playing games". If so, we can understand the need for clarity.
> > >
> > > Updated 3.1:
> > > We selected Chain-of-Thought [Wei et al., 2022b] prompting for its ubiquity and Reasoning-via-Planning [Hao et al., 2023] for its state-of-the-art status using Monte Carlo Tree Search (MCTS). MCTS has seen remarkable use for decision making in AI systems like AlphaGo Zero [2] and its successors and creating superhuman level capabilities within games like Go, Chess, Shogi, [3] among many others. [4]
> > >
> > > [2] Silver, D., Schrittwieser, J., Simonyan, K. _et al._ Mastering the game of Go without human knowledge. _Nature_ **550**, 354–359 (2017). https://doi.org/10.1038/nature24270
> > > [3] Silver, David, et al. "Mastering chess and shogi by self-play with a general reinforcement learning algorithm." _arXiv preprint arXiv:1712.01815_ (2017).
> > > [4] Browne, Cameron B., et al. "A survey of monte carlo tree search methods." _IEEE Transactions on Computational Intelligence and AI in games_ 4.1 (2012): 1-43.
> > >
> > >
> > > Updated related works:
> > >
> > > *LLM agents playing games*
> > > Recent work by [5,6] has also explored LLM capabilities in reasoning-heavy games, using games like Poker and Minesweeper. While these benchmarks focus on similar reasoning categories as we do, these papers focus on well-documented games with extensive online strategy resources, which could influence the agent's baseline strategic abilities. In contrast, our work focuses on lesser-known, multi-agent card games with minimal online presence, providing a more rigorous evaluation of strategic reasoning in OOD environments.
> > >
> > > [5] Giadikiaroglou, Panagiotis, et al. "Puzzle Solving using Reasoning of Large Language Models: A Survey." arXiv preprint arXiv:2402.11291 (2024).
> > > [6] Li, Yinghao, Haorui Wang, and Chao Zhang. "Assessing logical puzzle solving in large language models: Insights from a minesweeper case study." arXiv preprint arXiv:2311.07387 (2023).

---

### Official Review · Reviewer_K8mi · 2024-07-09
**Interesting benchmark; sparse evaluations**

**Rating:** 4
**Confidence:** 4

**Review:**

The authors have developed a very interesting benchmark of games (e.g. CodeNames) with associated environments for agents to play. While I do believe that this benchmark would be very useful and does show interesting reasoning problems, the evaluations are severely lacking. First many of the important details are relegated to the appendix (e.g. the main paper doesn't even have a one-line summary of the games). Each agent is sparsely evaluated against other agents making all the per-game findings inconclusive. For example, one of the findings is followed by statement (#153):
> But due to both GPT-4-RAP and the human baseline having very few data points, this detail should not be taken very seriously.

Once you look at the appendix, almost all agent pairs have played zero matches in most cases. I worry this caveat applies to all their findings. Even main tables (e.g. Table 3) have NaNs due to zero matches. Its hard for me to recommend an accept for a "Dataset & Benchmark" paper with such few data points being used to draw conclusions.

**Strengths:**

- A very interesting benchmark of games with associated environments to evaluate strategic reasoning ability of LLMs
- Strong agents with CoT and RAP scaffolding using SoTA LLMs

**Additional Feedback:**

See above

**Clarity:**

The paper is mostly well written. Some of the details from the appendix could be moved to the main paper if possible. E.g. brief game description, #matches/game.

**Correctness:**

The benchmark is likely constructed in a sound way. The Sea Battles result does make me worry a bit. Did the humans play using the same game environment as the agents? Were the humans asked/allowed to provide any bug reports?

**Documentation:**

Codebase was provided.

**Limitations:**

In addition to "Low Resolution Human Benchmark", it also seems that this is a low resolution agent benchmark too. I understand these experiments can be costly, but maybe couple of the strongest agents could be evaluated on few of the games a bit more thoroughly.

**Opportunities For Improvement:**

- While I understand the Bradley–Terry model doesn't require all agents to play each other, there are currently agents that have played no matches in certain games.
- Also is there some sense of error margins on these findings with so few matches? E.g. how much do the numbers deviate if you played 5 more matches per agent-pair?
- Table 3 should not have NaNs especially for the human baseline.
- #151
> The human baseline outperforms all model and scaffolding configurations in the benchmark

This is confusing. Table 2 (right above this line) clearly shows the agents outperforming humans on many games. If this is based on the aggregate metric, clarify that.

- Is Codenames the only multi-player game? Based on the appendix, that doesn't seem to be the case. So why do agents outperform humans only on the codenames game? Wouldn't the hypothesis that LLMs perform better "because they are better at modeling their teammate’s thought process" apply to other multi-player games?

- Its unclear why GPT-4 underperforms even random baseline on SeaBattle? This result makes me worry that something might be wrong in the game (or prompt) setup casting doubt on the other findings. Some examples of the matches with the GPT4 agents would be useful to mitigate such concerns.

- Some of the future work seems like should be part of this work, e.g. g-factor to handle SeaBattle, ideas in Sec 4.4.

**Relation To Prior Work:**

Somewhat related:
- BoardgameQA: A Dataset for Natural Language Reasoning with Contradictory Information
Mehran Kazemi, Quan Yuan, Deepti Bhatia, Najoung Kim, Xin Xu, Vaiva Imbrasaite, Deepak Ramachandran

**Summary And Contributions:**

The paper presents a new benchmark of strategic games to evaluate LLM's ability to perform strategic reasoning. Due to the nature of the games, this benchmark probes the LLM's ability to perform reasoning, communicate effectively, perform deductions, handle missing information, etc. Using the Bradley–Terry model, they show that GPT-3.5 and GPT-4 models even with COT-prompting and search (RAP) underperform humans on these games. The experiments compare GPT agents to humans on each game as well as in aggregate.

---

> ### Author Rebuttal · Authors · 2024-08-18
>
> We would like to thank the reviewer for their in-depth reading and analysis of the paper and useful feedback.
>
> R2 – “Its hard for me to recommend an accept for a "Dataset & Benchmark" paper with such few data points being used to draw conclusions” We would like to clarify that our primary contribution is the benchmark, which we list first among our contributions on line 55. We believe this is still on-scope for the Datasets and Benchmarks track based on the “Call for Datasets and Benchmarks” page on the NeurIPS website (“data generators and reinforcement learning environments”, “benchmarks on new or existing datasets, as well as benchmarking tools”)
>
> The data we include serves the purpose of demonstrating what insights could be learned from investing in this benchmark as opposed to only being able to discuss hypothetical results if one were to collect data.
>
> The data also helps to offset some of the costs for future, 3rd-party data collectors (as well as ourselves). Data collection for this benchmark was nearly prohibitively expensive for us due to funding constraints. On line 504 we offer a ballpark formula for the cost of running GPT-4-RAP as the cost of running the base GPT-4 model x the depth of the MCTS search x the number of actions available per game state x 6. This is high for us, but we believe that this is  reasonable for frontier labs. We agree that we do not have a high number of datapoints, but (1) our mass of data represents a substantial resource investment and (2) we believe our conclusions do not overextend given our limited data
>
> R2 – "I worry this caveat applies to all their findings”: We would like to clarify that this caveat is specifically meant for the detail that the confidence intervals for GPT-4-RAP and the human subject overlap. To elaborate: we wouldn’t interpret this to mean that GPT-4-RAP is near-human performance, rather to interpret it as both agents have rather wide CIs due to limited data. We wanted to avoid insinuating that scaffolded GPTs are now more strategically capable than humans.
>
> R2 – “there are currently agents that have played no matches in certain games”: GPT-4-RAP was very expensive to run. The human subject did not play all games because of time constraints. Depending on the game, one match could take over an hour, and our human subject couldn’t dedicate this much time in one sitting. We also believe that collecting more datapoints from one human subject is minimally informative. We believed that if we did not include any human data, this would be an obvious hole in our paper. The point of including human data was to very loosely ballpark human ability relative to frontier model ability, acknowledging that a much more robust and large-scale human data collection would have to be conducted in order to properly benchmark humans against frontier models. But no matter how many games we collected with our human subject, we still wouldn’t have enough data to generalize to the general human population’s strategic reasoning ability.
>
> R2 – “Table 2 (right above this line) clearly shows the agents outperforming humans on many games. If this is based on the aggregate metric, clarify that.”: You are right, we should clarify that the human outperforms the models based on the aggregate metric. Table 2 uses bold numbers to indicate the best score in each game, and underlined numbers to indicate the second-best. The human agent and GPT-4-RAP are tied for the most games with the highest score at 3 games. Then further, the human agent has second best score in 3 games, whereas GPT-4-RAP only has second-best score in two games. Finally, the “overall” column in Table 2 shows that the human agent has the highest score in the aggregate metric.
>
> R2 – “some sense of error margins on these findings with so few matches? ” We are a bit unclear with our understanding of this question. Figure 1b shows 90% confidence intervals for the overall agent rating. More matches per agent-pair would likely shrink these intervals assuming our current data is representative. These CIs were calculated using bootstrapping as described on line 133, meaning that we calculate ratings in hypothetical scenarios where more or fewer matches per agent-pair occurred.
>
> R2 – “humans play using the same game environment?”: All agents including the human agent used the same environment. Please see the attached PDF in the global rebuttal for a figure depicting example human gameplay. The only difference between the human agent and the GPT agents is the GPT agents were also fed various prompt scaffolds to help them interact with the game properly, whereas the human subject knew the rules and controls, and didn’t need the same support.
>
> R2 – “Were the humans asked/allowed to provide any bug reports?” – The human subject was not directly asked to provide bug reports, but they were certainly allowed to or fix them themselves, as the human subject also contributed to the codebase for implementations of several games. If not fixing directly, we communicated frequently over Slack during data collection and had let us know when other issues were arising.
>
> R2 – “Table 3 should not have NaNs especially for the human baseline.” – We use NaNs to indicate the games that the human did not play, instead of 0s which might suggest that the human played but did poorly.
>
> R2 – “BoardgameQA”; We agree that this paper would have been a good addition to our related works section. This paper studying LLM reasoning ability with contradictory information seems related to our work, and more pertinently their focus on “Incomplete Information” in the dataset creation is highly applicable.
>
> R2 – “Many of the important details are relegated to the appendix” Thank you for this feedback as this is something we were undecided on. We were worried that the specifics about each game would break the main narrative of the paper (especially if we had more games), and chose to describe broadly the themes of the games starting in Table 1.

---

### Official Review · Reviewer_bYEh · 2024-07-30
**Need to clarify whether you implement the games and add sufficient example of the games.**

**Rating:** 6
**Confidence:** 4
**Correctness:** The claims made in the submission are…

**Review:**

The quality of the work is solid, presenting a well-defined benchmark and thorough evaluation of LLMs in strategic reasoning tasks.

The clarity is good. But important details on implementing the games, as well as sufficient examples showcasing how these games look like is needed.

The originality lies in the cross-domain benchmark focusing on out-of-distribution strategic reasoning.

The significance is notable as it addresses the gap in evaluating LLMs in complex strategic scenarios, which is crucial for developing more capable AI systems.

**Strengths:**

1. The paper introduces a unique benchmark for evaluating strategic reasoning in LLMs, filling a crucial gap in the field.
2. Detailed analysis of model performances and case study on models' sensitivity are interesting.
3. The paper covers a wide range of strategic reasoning tasks across diverse game environments.

**Additional Feedback:**

1. The paper contains a Acknowledgment section which seems nullify the anonymity.
2. I can't open the website at the moment I write this review:  https://anonymous.4open.science/r/GameBench-5942/

**Clarity:**

The paper is well written and easy to follow. One thing can be improved is the overview of the method. I still not sure if the authors implement all these games in python code or they use python API to call some game softwares.

**Documentation:**

I can't open the code repo at this moment so I can't assess the documentation.

**Ethics:**

No ethical concern at all. I appreciate the authors for detailing their research on human subjects in Appendix B.

**Limitations:**

The authors adequately addressed the limitations.

**Opportunities For Improvement:**

1. I am not sure if the authors implement these games in python code and create environment to evaluate agents or they just call some existing APIs. This affects the evaluation of the contribution of this work.
2. More examples on the games are needed to help reader form a concept on what the paper is doing. Full example on human's game-playing is needed so we can understand why in Table 3 human has such a high score.

**Relation To Prior Work:**

The paper discussed prior work thoroughly. The paper clearly differentiates its contributions from previous benchmarks by focusing on out-of-distribution strategic reasoning and providing a comprehensive multi-game evaluation.

**Summary And Contributions:**

The submission presents GAMEBENCH, a comprehensive benchmark for assessing the strategic reasoning skills of LLM agents across nine different game environments, each representing distinct reasoning challenges found in strategy games. These games are mostly text-based card games. **However, I am not sure if the authors implement these games in python code and create environment to evaluate agents or they just call some existing APIs.**


The authors employ GPT-3 and GPT-4 models and evaluate the impact of Chain-of-Thought (CoT) prompting and Reasoning Via Planning (RAP) scaffolding frameworks.

The results indicate that neither model matches human performance, with GPT-4 performing worse than random actions in some scenarios. CoT and RAP scaffolding improve performance but still fall short of human levels.

---

> ### Author Rebuttal · Authors · 2024-08-18
>
> We thank Reviewer bYEh29 for the helpful feedback, which we have incorporated and believe that the final message of the paper is stronger as a result. Please see our detailed responses below.
>
> R1 – “I am not sure if the authors implement these games in python code”: On line 121 we write that each environment is written in Python, but we agree that this could be more specific. We ourselves wrote the Python implementations, and they are available in our open-source repository. The only API we used was OpenAI’s API for interacting with its models, which we mention on line 501.
>
> R1 – “More examples on the games are needed... as well as sufficient examples showcasing how these games look like is needed… Full example on human's game-playing is needed so we can understand why in Table 3 human has such a high score”: Please see the PDF in the global rebuttal for a figure demonstrating what an instance of gameplay looked like to the human player. We describe each game starting on line 570. Gameplay for each GPT-based agent looks like a conversation over the OpenAI API. Gameplay for the human agent looks similar except this conversation takes place over a terminal or command prompt, and there is less prompt scaffolding since we don’t need to tell our human player to reason out loud, for example.
>
> R1 – “I can't open the code repo at this moment so I can't assess the documentation”: Thank you for letting us know. One of our authors faced this issue once as well and they were able to solve it by refreshing the page. We suspect it could be because of their internet service provider, so perhaps a VPN could additionally help.
>
> R1 – “The paper contains an Acknowledgment section which seems nullify the anonymity”: This PDF was generated using the NeurIPS template for Datasets and Benchmarks with the “anonymous” option enabled. We agreed it was peculiar that this did not hide the acknowledgements section like it hides the authors, but decided this was likely intentional. Further, there was nothing in the style guide that would indicate to us that we needed to manually hide this section.

---

### Author Rebuttal · Authors · 2024-08-18

We would like to thank all of the reviewers for their time and effort in their reviews. The attached PDF has screenshots of human gameplay in a Windows command prompt for the reviews that asked for it.

---

### Decision · Program_Chairs · 2024-09-26

**Decision:**

Reject

**Comment:**

This paper presents a new benchmark of strategic games to evaluate LLM's ability to perform strategic reasoning. After rebuttal, it received scores of 346. All the reviewers had some shared concerns: (1) paper writing needs improvement, e.g., the games themselves are not even introduced in the main text; (2) the evaluations are severely lacking and the analysis of results is insufficient; (3) some highly related works are missing. Overall, the AC would like to recommend rejection of the paper.